# High-Linearity Hydrogel-Based Capacitive Sensor Based on *Con A*–Sugar Affinity and Low-Melting-Point Metal

**DOI:** 10.3390/polym14204302

**Published:** 2022-10-13

**Authors:** Ruixue Yin, Jizhong Xin, Dasheng Yang, Yang Gao, Hongbo Zhang, Zhiqin Qian, Wenjun Zhang

**Affiliations:** 1School of Mechanical and Power Engineering, East China University of Science and Technology, Shanghai 200237, China; 2Shanghai Key Laboratory of Intelligent Sensing and Detection Technology, East China University of Science and Technology, 130 Meilong Road, Shanghai 200237, China; 3Division of Biomedical Engineering, University of Saskatchewan, 57 Campus Drive, Saskatoon, SK S7N 5A9, Canada

**Keywords:** hydrogel, glucose capacitive sensor, low-melting-point metal, serpentine electrodes, continuous glucose monitoring system

## Abstract

Continuous glucose monitoring (CGM) plays an important role in the treatment of diabetes. Affinity sensing based on the principle of reversible binding to glucose does not produce intermediates, and the specificity of concanavalin A (*Con A*) to glucose molecules helps to improve the anti-interference performance and long-term stability of CGM sensors. However, these affinity glucose sensors have some limitations in their linearity with a large detection range, and stable attachment of hydrogels to sensor electrodes is also challenging. In this study, a capacitive glucose sensor with high linearity and a wide detection range was proposed based on a glucose-responsive *DexG–Con A* hydrogel and a serpentine coplanar electrode made from a low-melting-point metal. The results show that within the glucose concentration range of 0–20 mM, the sensor can achieve high linearity (R^2^ = 0.94), with a sensitivity of 33.3 pF mM^−1^, and even with the larger glucose concentration range of 0–30 mM the sensor can achieve good linearity (R^2^ = 0.84). The sensor also shows resistance to disturbances of small molecules, good reversibility, and long-term stability. Due to its low cost, wide detection range, high linearity, good sensitivity, and biocompatibility, the sensor is expected to be used in the field of continuous monitoring of blood glucose.

## 1. Introduction

Diabetes is one of the leading causes of death. The number of people suffering from diabetes is more than 422 million worldwide, and the number of people dying from diabetes approaches 1.6 million per year [1]. Diabetes is a metabolic disease characterized by high glucose concentrations in the blood [2]. Measuring blood glucose concentration is the most important step in treating diabetes. The most widely used whole-blood sampling can only measure the blood glucose concentration at one timepoint, making it unable to identify hypo/hyperglycemia in real time [3]. Glucose sensors capable of performing continuous glucose monitoring can provide real-time information about blood glucose [4] and are attracting more and more attention in current research.

To date, there are two main types of glucose sensor: electrochemical sensors based on the glucose oxidation principle [5,6,7,8,9], and affinity sensors from reversible binding with glucose [10]. Since Clark and Lyons first proposed an electrochemical sensor based on the glucose oxidase (GOx)-catalyzed glucose oxidation reaction [11], electrochemical sensors have become the most widely used blood glucose sensors, with the advantages of high sensitivity, good repeatability, easy manufacture, and relatively low cost [12]. However, this enzymatic reaction is affected by the concentration of humoral oxygen [13]. Although oxygen can be replaced by a synthetic electron redox mediator, and the intermediate medium can even be removed by modifying the electrode with nanostructures [14], this kind of sensor is still limited by the dependence of enzyme activity on temperature, humidity, and interference, along with its short life expectancy of no more than two weeks in CGM applications. In recent years, researchers have used metals (such as Pt, Au, Ni, Cu, Pd, and metal oxides such as CuO, NiO, etc.) as catalysts instead of GOx to catalyze the oxidation of glucose [15,16,17,18,19,20]; however, the manufacturing of these sensors requires a complex fabrication process, and it is hard to avoid the formation of intermediate products in the electrochemical reaction and their attachment to the electrode surface, resulting in a poisoning effect [21]. Compared with electrochemical sensors, glucose sensors that rely on affinity elements—such as phenylboronic acid (PBA) [22,23,24,25] and concanavalin A (*Con A*) [26,27,28,29,30]—do not produce intermediates. The affinity sensors produce signal changes based on the principle of specific and reversible binding with glucose. Among them, Con-A-based affinity sensors have strong specific binding ability with glucose, offering great potential in the application of continuous glucose monitoring.

Ballerstadt et al. [31] proposed a fluorescent sensor for detecting glucose concentrations in the range of 0.2 to 30 mM based on *Con A*, but the sensor had insufficient long-term stability due to the leakage of *Con A* [32]. Chen [33] used a fluorescence resonance energy transfer (FRET) system to detect glucose concentrations, but the linear detection range was only 0.4–1.87 μM. Yu [34] incubated *Con A* on an electrode to detect the non-enzymatic electrochemiluminescence (ECL) intensity change caused by the competitive reaction between glucose and phenoxy-glucan, which could achieve the monitoring of glucose concentrations in the linear range of 1.0 × 10^−10^∼5.2 × 10^−5^ M. Cai [35] proposed a capacitive hydrogel sensor for detecting the change in capacitance of a glycidyl methacrylate dextran (*DexG*)–*Con A* hydrogel, achieving repeatable and reversible response in the glucose concentration range of 0–30 mM, but the coefficient of determination (R^2^) of linear fitting at 0–30 mM was only 0.54. At present, for *Con A* affinity sensors to achieve blood glucose monitoring with a wide detection range and good linearity is still challenging.

In this study, a capacitive glucose sensor with high linearity and a wide detection range was developed based on a glucose-responsive *DexG–Con A* hydrogel and serpentine coplanar electrode. The dielectric property of the *DexG–Con A* hydrogel changes accordingly with the change in environmental glucose concentration. An easily machined low-melting-point alloy (In_32.5_Sn_51_Bi_16.5_ alloy at the melting point of 62 °C) was used to fabricate the serpentine electrode (SRE) via a microfluidic method, which may significantly contribute to improving the hydrogel–electrode interface. A layer of PDMS and a semipermeable membrane were used to encapsulate the *DexG–Con A* hydrogel and act as a scaffold. The semipermeable membrane can allow the exchange of glucose between the sensor and the environment, protect the sensitive substances (e.g., *Con A* molecules) from loss, and block the passage of macromolecular disruptors from the environment. The experimental results show that compared with interdigital carbon electrodes, an SRE prepared from a low-melting-point alloy can provide better adhesion with hydrogels, and the encapsulation of the hydrogel also contributes to the high linearity of the proposed sensor.

## 2. Materials and Methods

### 2.1. Materials

InSnBi alloy was supplied by Shenyang Jiabei Commerce Ltd. (Shenyang, China). A PDMS elastomer kit (Sylgard 184) was purchased from Dow Corning (Midland, MI, USA). Concanavalin A (*Con A*, extracted from Jack Bean, pre-activated, M_w_ = 102 kDa, containing Ca^2+^ and Mn^2+^) was purchased from Medicago Inc. (Quebec City, QB, Canada). Polyethylene glycol (600) dimethacrylate (PEGDMA) was obtained from TCI, Inc. (Bellevue, WA, USA). Dextran (M_w_ = 70 kDa), glycidyl methacrylate (GMA), lithium phenyl-2,4,6-trimethylbenzoylphosphinate (LAP), phosphate-buffered saline (PBS), HCl, D-glucose, D-fructose, D-galactose, and ascorbic acid were purchased from Sigma-Aldrich (St. Louis, MO, USA). All of the regents were used as received. *DexG* was synthesized via the ring-opening reaction of GMA with dextran as described in our previous study [36]. The obtained derivative with 20.6% substitution (*DexG* T 70–20%) was used in the subsequent experiments.

### 2.2. SRE Fabrication

In this study, the liquid–solid phase transition of a low-melting-point alloy in microfluidic channels was used to fabricate the SRE. The manufacturing process is shown in Figure 1A. First, the microfluidic channels were formed by curing PDMS in a mold made by a 3D printer (B9creations, Rapid City, SD, USA). Then, the cured PDMS microfluidic channels and glass slides were treated in an oxygen plasma treatment machine (Chengdu Mingheng Technology Development Co., Ltd., Chengdu, China) for 4 min and then combined together. Next, the low-melting-point alloy was injected into the microfluidic channels in an oil bath at a temperature of 85 °C, and the electrode was obtained after the low-melting- point alloy was cooled and transformed.

### 2.3. Fabrication and Packaging of the Capacitive Glucose Sensor

The glucose-responsive *DexG–Con A* hydrogel was formed in situ on the surface of the SRE, and then PDMS was used to fix the semipermeable membrane on the electrode to package the sensor. Briefly, we dissolved 100 mg mL^−1^ of *DexG* and 23 mg mL^−1^ of *Con A* in buffer (PBS, pH 7.4), and then used SBL ultrasound (Ningbo Scientz Biotechnology Co., Ltd., Ningbo, China) with 30% power ultrasonic vibration for 2 min. After that, the mixture was left to stand for 3 h to fully crosslink the *DexG* and *Con A*. Then, 12.3 mg mL^−1^ of PEGDMA and 1 mg mL^−1^ of LAP photoinitiator were added and vibrated for 2 min. Finally, 40 μL of the mixture was transferred to the electrode surface in the PDMS chamber (7 mm × 7 mm × 0.1 mm), treated with oxygen plasma, and irradiated with 365 nm UV light at 106 mW light intensity for about 10 s.

The procedure of packaging was to fix the semipermeable membrane on the sensing element and then isolate the sensing element from the external solution to be tested. Briefly, the treated dialysis bag (Beijing Solarbio Science & Technology Co., Ltd., Beijing, China) was cut to an appropriate size and assembled on the electrode substrate together with the packaging parts made of PDMS. After encapsulation, the PBS buffer solution was injected into the cavity with a 1 mL syringe until the cavity was full. The obtained sensor and its detailed structure are shown in Figure 1B,C.

### 2.4. Characterization 

In order to verify the yield of electrode fabrication, 12 interdigital electrodes (IDEs) and SREs were made by the microchannel method, and their resulting yields were compared. Moreover, different proportions of PDMS were used to make flow channels to form the SREs. The air capacitance of the obtained electrodes was measured and compared using an LCR meter (IM3536 LCR METER HIOKI) to determine the appropriate proportion of PDMS. Furthermore, to investigate the electrode–hydrogel interface, the cross-section of the connection between the hydrogel and the electrode was observed with inverted microscopes (Nikon Instruments Inc., Melville, NY, USA).

### 2.5. Performance Rests of the Capacitive Glucose Sensor

The capacitance of the sensor was measured using an LCR meter, and the applied voltage was V_AC_ = 5 V. A wire was used to connect the end of the SRE and the detector. PBS buffer solution (pH 7.4) was used to dissolve the glucose for the tests. The concentration of glucose ranged from 0 mM to 30 mM. The capacitance change caused by the change in glucose concentration was continuously measured at 30 kHz in the same way. In addition, ascorbic acid, galactose, and fructose were added to 5 mM glucose buffer with a glucose concentration of 10% to measure the capacitance change of the sensor under interference. The performance test of the sensor in this study was performed in a fluid chamber of the in vitro test platform. A spiral pump was used to exchange the glucose solution, with the pump flow set at 20 mL min^−1^ and the flow rate at 5.56 cm min^−1^, which is within the flow rate range of tissue fluid in the human body (0–7 cm min^−1^). The time of each fluid change was about 30 s, which could be ignored in the process of the experiment. For the long-term stability test, the sensor was transferred from the fluid chamber to PBS (pH 7.4) and stored at 4 °C, and then it was restored to room temperature and measured in the same environment when needed.

#### Cytotoxicity Test

The CCK-8 kit was used to detect the in vitro cytotoxicity of the InSnBi alloy electrode and glucose sensor proposed in this study. Before preparation of the extract, the InSnBi electrode or sensor was impregnated with absolute ethanol for 24 h to complete the bactericidal treatment. Then, the extract was prepared according to the GB/T 16886 standard, and one sterilized InSnBi electrode or sensor was immersed in DMEM medium (pH 7.4) for 72 h. C2C12 cells were then cultured with the InSnBi electrode or sensor extract and DMEM medium. The pH value for the cell culture was 7.4. Cells cultured with the extract of the InSnBi electrode or sensor were used as the experimental group, and cells cultured with DMEM medium were used as the control group. After the cells were incubated for 1–3 days, 10 μL of CCK-8 reagent was added to the cells. After 2 h of incubation, the absorbance at 450 nm was measured using a microplate reader, and then the number of viable cells was calculated. 

## 3. Results and Discussion

### 3.1. Fabrication and Optimization of the Coplanar Electrode

Low-melting-point alloys can be used to easily fabricate complex shapes. Direct ink writing, atomization spraying, 3D printing, acoustic printing, and microfluidic methods can be used to process low-melting-point metals to produce a variety of structures. In this study, we used a microfluidic method to achieve low-cost manufacturing with the desired coplanar structure (see Figure 1A). The melting point of the InSnBi alloy used in this study is 62 °C—much higher than the normal temperature of the human body—which can ensure its safety for in vivo use as electrodes. Most importantly, InSnBi is biocompatible, while also meeting the requirements of easy processing [37]. 

Coplanar electrodes are widely used in nondestructive testing sensors and pressure, electrical, and acoustic transducers. Their structural characteristics enable them to sense on one side, and they can simply change their size to control the signal strength. The shape of coplanar electrodes can be interdigital, meander, spiral, serpentine, etc. Among them, the serpentine structure and the interdigitated structure have higher capacitance per unit area [38], generally leading to higher sensitivity of corresponsive capacitive sensors. Therefore, in this study, SREs and IDEs were compared in terms of their manufacturing yield to determine the most suitable electrode structure. The results show that the preliminary manufacturing yield of IREs is about 42%, while that of SREs is about 75%. One possible reason for this phenomenon is that the number of outlets of the channel for molten metal injection of the interdigitated structure is four (Figure 2A), while the number of outlets of the serpentine structure is only two. The increase of the number of outlets greatly increases the difficulty of operation during metal injection. When the metal is injected and waiting for the alloy to cool, the shrinkage caused by the shrinkage of the alloy is also more serious for more outlets. Considering its higher manufacturing yield and high capacitance, the serpentine structure was deemed to be more suitable as the electrode structure for the glucose sensor in this study.

In addition, we should note that the hardness of PDMS microchannels prepared using different proportions of the PDMS base and the curing agent also differs, leading to different deformation of the flow channel when injecting the low-melting-point metal [39], and resulting in differences in the metal area in the electrode. The metal area is a key factor affecting the electric capacity [40] and the sensitivity of the obtained capacitive sensor. Therefore, we investigated PDMS with different proportions to make flow channels to fabricate SREs, and we measured the air capacitance for the optimization of electrode fabrication. The results in Figure 2B show that the lower the content of the curing agent, the greater the capacitance of the electrode and its error. The reason for this may be that the lower the content of the curing agent, the greater the flexibility of PDMS, and this flexibility makes the microchannel easy to deform under pressure and disturbance. The higher capacitance and higher error of the SRE electrode with less curing agent in the PDMS channel are due to its greater flexibility, deformation, and uncertainty. The air capacitance of the SRE manufactured by the PDMS channel with a 5:1 ratio of base to curing agent of was as much as 98.45% of that of the 10:1 ratio, while the error was only 37.22% compared with the error of the 10:1 group. Therefore, PDMS with a 5:1 ratio was selected to fabricate an SRE with higher capacitance as well as lower error.

### 3.2. Characterization of the Electrode–Hydrogel Interface

The electrode–hydrogel interface plays an important role in the performance of hydrogel-based sensors. Myoung Seon covalently connected a humidity-sensitive polymer with an electrode to improve the stability of the sensor [41]. For the *DexG–Con A* hydrogel-based glucose sensor developed in our group, an equilibrium of glucose (*Glu*), *DexG*, and *Con A* binding existed in the hydrogel, and the equilibrium equation can be derived from ligand competition theory [26,42]:(1)ConA+DexG⇌ConA−DexG
(2)ConA+Glu⇌ConA−Glu

When the concentration of free glucose changes, the degree of crosslinking of the *Con A–DexG* network changes as the equilibrium is broken, and the volume of hydrogel increases or decreases, which eventually affects the stress distribution between the hydrogel and the electrode. When the hydrogel is not sufficiently attached on the substrate, the solution in the environment will inevitably penetrate into the gap between the hydrogel and the substrate, resulting in an unnecessary change in capacitance. In our previous study, we proposed a capacitive glucose sensor made from a *DexG–Con A* hydrogel formed in situ on the surface of a carbon electrode prepared by direct laser writing [35]. It was observed that there was a gap at the interface between the hydrogel and the carbon electrode, as shown in Figure 2C. This is because during the direct laser writing process, after the C-H, C-O, and other bonds in the material are broken, the atoms quickly overflow in the form of H_2_, CO_2_, and other gases, resulting in the formation of a loose porous structure of the carbon electrode. Therefore, air molecules are easily trapped in the pores, enhancing the hydrophobicity of the electrode [43], and leading to the gap at the interface. Then, the gap may decrease the adhesion of the hydrogel on the carbon electrode. The adhesion of the hydrogel may not be able to offset the stress changes caused by swelling, resulting in the exfoliation of hydrogels from carbon electrodes, which may increase the failure rate of sensor fabrication. Conversely, in this study, the use of a low-melting-point metal with good hydrophilicity and excellent conductivity instead of a porous carbon structure improved the adhesion of the hydrogel on the surface of the electrode. From Figure 2D, it can be seen that even when cutting, the metal electrode and the hydrogel are very closely connected.

### 3.3. Performance of the Capacitive Glucose Sensor

The dielectric measurement of coplanar electrode depends on the combination of the application of spatial periodic potential on the surface of the object to be measured and the change in the excited electric field in the medium, resulting in the change in the dielectric coefficient in the electric field of the capacitor. The volume of the *DexG–Con A* glucose-responsive hydrogel changes with the concentration of the glucose solution, and its dielectric properties also change accordingly [26]. Therefore, theoretically measuring the capacitance change of the hydrogel can determine the change in glucose concentration in its environment.

In order to study the performance of the sensor more objectively, we tested four capacitive glucose sensors at the same time and normalized the measurement results. The normalization formula was obtained through Formula (3), where $C_{i}$, $C_{min}$, and $C_{max}$ are the capacitance response under the current concentration, the minimum capacitance response, and the maximum capacitance response, respectively:(3)$$N=\frac{C_{i}-C_{\min}}{C_{\max}-C_{\min}}$$

Figure 3A shows the response of the proposed sensor under different glucose concentrations. It should be noted here that the N value represents the results after normalization. The real capacitance is shown in Figure 3D. Considering that the extreme hyperglycemia of a diabetic patient is higher than 22 mM [44], a 0–30 mM glucose concentration range was tested. The large error bar may have been caused by the experimental error during the fabrication and measurement of the four different samples, as well as the interferences from the environment and the electrode connection. In real applications after sensor packaging, the Y-error should be much smaller. The capacitance of the sensor decreases monotonically when the glucose concentration increases from 0 mM to 30 mM, and the capacitance response of the sensor within this range shows a good linearity with the change in glucose concentration (R^2^ = 0.84). Within the glucose concentration range of 0–20 mM, the sensor can achieve higher linearity (R^2^ = 0.94). In Cai’s study [35], the linearity between the capacitance response of the sensor and the change in glucose in the concentration range of 0–30 mM was only R^2^ = 0.54; the sensitivity of the sensor decreased seriously when the glucose concentration increased. It was also found that the capacitance response decreased disproportionally, with great changes at glucose concentrations of 3 and 6 mM. This may be related to the swelling mechanism of the *DexG–Con A* hydrogel [26,35,36]. The swelling of the *DexG–Con A* hydrogel occurred rapidly at first, and then continued more gradually until reaching equilibrium, owing to the reduced available binding sites on *Con A* at increased glucose concentrations. Figure 3B shows that the capacitance response of the SRE does not show a monotonic property with the change in glucose concentration, indicating that the capacitance response of the glucose sensor is independent of the inherent properties of the SRE. The capacitance is positively correlated with the dielectric constant between electrodes. 

To further investigate the factors for linearity, the capacitance of a sensor without encapsulation by a semipermeable membrane was measured, as shown in Figure 3C. Similarly to Figure 3A, the capacitance showed a monotonic response to the change in glucose concentration within 0–6 mM, but with a greater rate of decrease compared with Figure 3A. Meanwhile, the capacitance change became mild at higher glucose concentrations—in particular, the N values at 6, 12, 15, and 21 mM appeared similar. The use of the semipermeable membrane may slow the binding of glucose with *Con A*, slowing the swelling of the *DexG–Con A* hydrogel and improving the linearity of the sensor. At the same time, there may be leakage of *Con A* without encapsulation by the semipermeable membrane. The interaction of these two possible phenomena may have contributed to the results shown in Figure 3C.

In addition, the sensitivity of the sensor was defined and calculated using Equation (4), where ΔC and ΔGlu are capacitance difference and the concentration difference, respectively:(4)$$Sensitivity=\frac{\Delta C}{\Delta Glu}$$

The result shown in Figure 3D is a complete response record of the proposed glucose sensor, showing that the sensitivity of the glucose sensor can reach 24 pF mM^−1^ in the glucose concentration range of 0–30 mM. Within the glucose concentration range of 0–20 mM, the sensor can achieve higher sensitivity of 33.3 pF mM^−1^. Compared with other capacitive glucose sensors [45,46], the proposed sensor in this study shows a wider range of detection ability, higher sensitivity, and better suitability for the continuous monitoring of glucose in the body.

The components contained in the in vivo environment are more complex than the experimental environment in vitro. There are a large number of interfering substances (ascorbic acid, galactose, fructose, etc.) in the body, which may interfere with the capacitive response of the sensor. In order to study the influence of interfering substances on the capacitive glucose sensor, we added 10% interfering substances to a 5 mM glucose solution for testing, considering that the physiological concentrations of these substances are about one order of magnitude lower than the glucose level. The results are shown in Figure 4A, and the capacitance response ratio of interference is determined by the capacitance ratio $C_{d}/C_{g}$, where $C_{g}$ is the capacitance response at 5 mM glucose concentration, while $C_{d}$ represents the capacitance response in the presence of interfering substances. The experimental results show that the capacitance response ratios of ascorbic acid, galactose, and fructose are 0.994, 0.998, and 0.995, respectively. This indicates that these substances have little effect on the selectivity of the proposed glucose sensor. We also measured the reversibility of the sensor to the glucose concentration by repeated changes of two glucose concentrations between 0 and 30 mM, and the results are shown in Figure 4B. Here, the capacitance response of the first cycle was selected as the basis for normalization. Finally, the normalized value at the concentration of 0 mM in the seven cycles was between 0.755–1.000 and −0.014–0.141 at the concentration of 30 mM. Therefore, we believe that the glucose sensor has good reversibility. Moreover, the long-term stability of the sensor is very important for its application in vivo. There was no chemical reaction, no material loss, and no intermediate product generation in the detection process of the proposed glucose sensor; hence, the long-term stability of the sensor is good in theory. We tested the capacitance response of the sensor for 7 days to preliminarily validate the long-term stability of the sensor, and the results are shown in Figure 4C. By normalizing the measurement results on day 1 and day 7, the drift rates of the sensor at 3 mM and 6 mM were found to be 2.5% and 6.6%, respectively. The 7-day drift rate of the carbon electrode sensor made by Cai et al. [35] based on a *Con A* hydrogel was 10.9%, and the drift rate of the electrochemical glucose sensor based on GOx made by Wang et al. [47] was 18.7%. The above results indicate that the Con-A-based glucose sensor prepared in this study has good stability. For future in vivo testing, a microneedle array is being developed to be incorporated with the sensor for the extraction of bodily fluids under the skin. The semipermeable membrane used in the encapsulation of the sensor can protect the sensor from the influence of biomacromolecules in the bodily fluids. 

### 3.4. In Vitro Cytotoxicity

In order to verify whether the proposed electrode and sensor are suitable for biomedical applications, the InSnBi electrode and sensor were tested for cytotoxicity in vitro using C2C12 cells with a pH 7.4 cell culture environment in DMEM. The experimental results are shown in Figure 5. Figure 5A shows that there was no significant difference in the proliferation rates of cells incubated with the InSnBi alloy extract and DMEM medium (Ctrl), indicating that the InSnBi alloy has no cytotoxicity, which is consistent with the results of previous studies [39]. Figure 5B shows that there was also no significant difference in the proliferation rates of cells incubated with the sensor extract and DMEM medium (Ctrl), indicating that the sensor was also not cytotoxic. In addition, the InSnBi alloy extract and sensor extract showed that with the extension of incubation time, the proliferation of incubated cells became better and better. These results indicate that the InSnBi electrode and the proposed sensor have good potential for biomedical applications.

## 4. Conclusions

In this study, a capacitive glucose sensor with high linearity and a wide detection range was proposed based on a glucose-responsive *DexG–Con A* hydrogel and serpentine coplanar electrode made from a low-melting-point alloy. PDMS microfluidic channels were used to form SREs from InSnBi alloy, with low cost and rapid manufacturing ability. Compared with interdigital carbon electrodes, SREs prepared from low-melting-point alloys provided better adhesion with the *–Con A* hydrogel. The encapsulation of the hydrogel by a semipermeable membrane also contributed to the high linearity of the proposed sensor. Eventually, the capacitive glucose sensor achieved a good linear response within the glucose concentration range of 0–20 mM, achieving high linearity (R^2^ = 0.94) with sensitivity of 33.3 pF mM^−1^. In general, the capacitive glucose sensor has the potential to be applied in the field of continuous glucose monitoring.

## Figures and Tables

**Figure 1 polymers-14-04302-f001:**
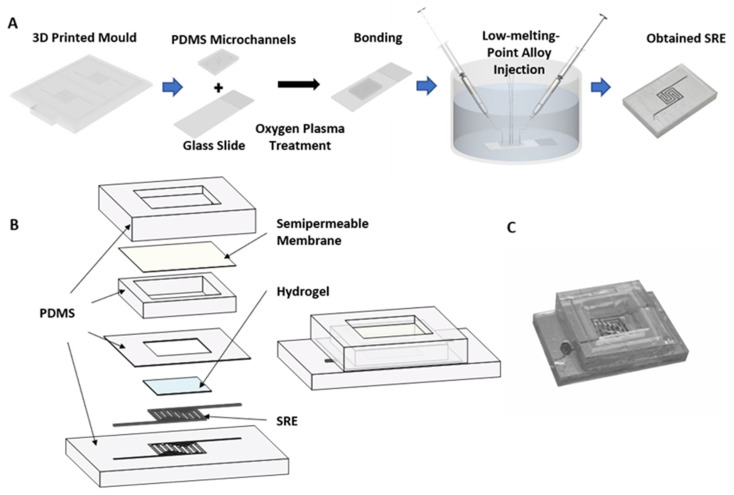
(**A**) Schematic illustration of the fabrication process of the SRE. (**B**) The exploded view and (**C**) the entity of the proposed capacitive glucose sensor.

**Figure 2 polymers-14-04302-f002:**
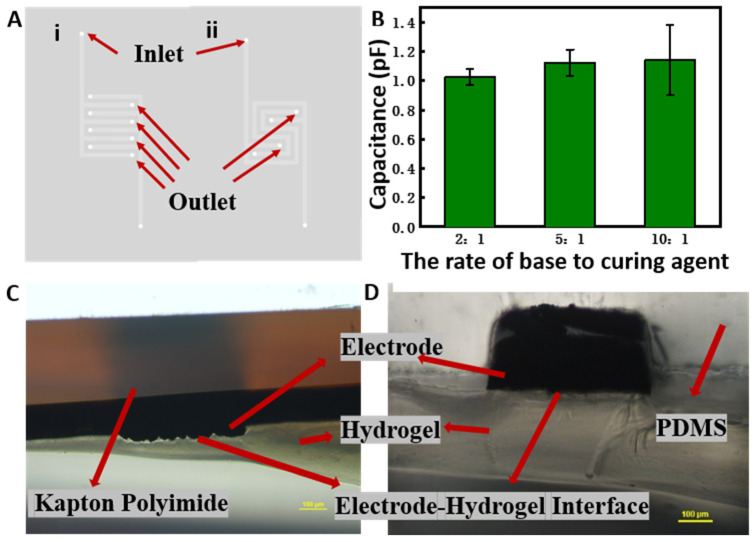
(**A**) Schematic illustration of fabricated interdigital and serpentine electrodes with PDMS microfluidic channels. (**B**) Air capacitance of an SRE fabricated using PDMS microfluidic channels with different proportions of base and curing agent, where the error bars reflect standard errors. (**C**) Cross-section of the interface between the direct laser writing carbon electrode and the hydrogel. (**D**) Cross-section of the interface between the low-melting-point metal electrode and the hydrogel.

**Figure 3 polymers-14-04302-f003:**
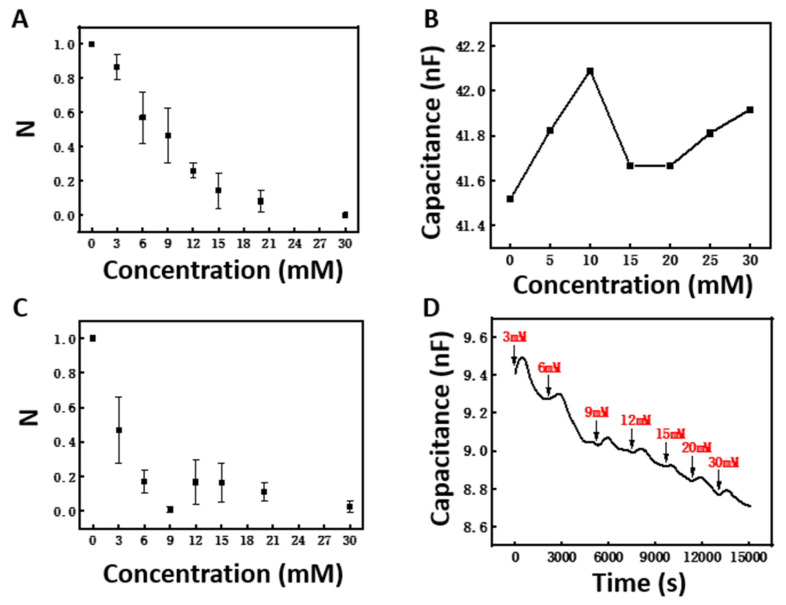
(**A**) Dependence of the capacitance of the capacitive glucose sensor on the glucose buffer solution (pH 7.4) with different concentrations at 30 kHz. (**B**) Dependence of the capacitance of the SRE without hydrogel integration on the glucose buffer solution (pH 7.4) with different concentrations at 30 kHz. (**C**) Dependence of the capacitance of the sensor without a semipermeable membrane on the glucose buffer solution (pH 7.4) with different concentrations at 30 kHz. (**D**) Continuous response curve of the capacitive glucose sensor at glucose concentrations ranging from 0 to 30 mM.

**Figure 4 polymers-14-04302-f004:**
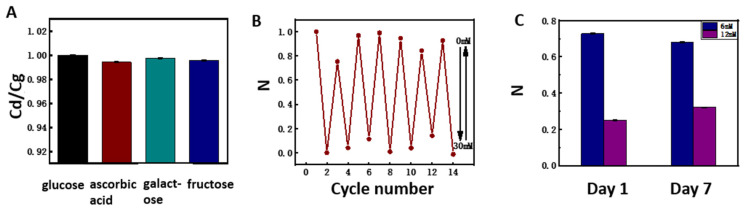
(**A**) Capacitance in response to disturbances in 5 mM glucose buffer solution. The disturbances include galactose, ascorbic acid, and fructose at concentrations of 0.5 mM. (**B**) Reversible capacitance response of the *DexG–Con A* hydrogel sensor immersed in glucose solution with repeated concentration changes between 0 and 30 mM at 30 kHz. (**C**) Stability test of the capacitive glucose sensor.

**Figure 5 polymers-14-04302-f005:**
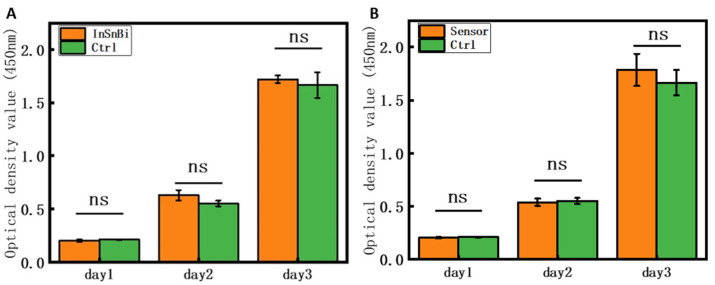
(**A**) The CCK-8 results of InSnBi extraction vs. negative control and (**B**) sensor extraction vs. negative control; ns: no significant difference.

## Data Availability

The data presented in this study are available on request from the corresponding authors.

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
