# Peer review of "High-Linearity Hydrogel-Based Capacitive Sensor Based on Con A–Sugar Affinity and Low-Melting-Point Metal"

_polymers, 2022, doi:10.3390/polym14204302_

Round 1

Reviewer 1 Report

This is an interesting experimental work on making capacitance sensing of glucose. The emphasis is on the linearity of the sensor. Con A hydrogel and low melting temperature metallic alloys were impregnated into the 3D printed framework for building the sensor. The paper delivers some new information in view of the fabrication and the characterization of the sensing performance.

Some minor touch may be made before publishing.

1. The authors' affiliation followed by complete addresses should be added. Fir example, including zip code or mail code may be useful for contacting.

2. Discussion on possible in-vivo testing or plan for such experiment may be added. It is recommended that the discussion on the reliability of in-vitro test to predict or validate in-vivo results may be mentioned.

3. Is the alloy In-Bi-Sn alloy stable in body fluids? How about its toxicity?

In summary, this is a nice work on application oriented work. It is acceptable after minor revisions. 

Reviewer 2 Report

The authors propose based on glucose-responsive DexG-Con A hydrogel and serpentine coplanar electrode made by low-melting-point metal, and the results show that the sensor can achieve high linearity (R2=0.94) and a sensitivity as low as 33.3 pF mM-1. The results are interesting and will be of interest to researchers in the field of polymer networks and electrochemical sensor. However, some methods and data descriptions in the manuscript are unclear. I recommend publication, if the following mandatory revisions are made well. Below are some specific comments:

1.    Page 1, Abstract: The addition of the glucose detection range and quantitative results to the abstract highlights the manuscript quality of the study.

2.    The authors state that the semipermeable membrane is immobilized on the sensing element, but does not clearly state the function of using the semipermeable membrane (a scaffold for the hydrogel DexG-Con A, dialysis, or something else)?

3.    Figure. 2 (A)Figure. 3 (Coordinate title) Figure. 4 (Coordinate title) has weak quality.  Please improve.

4.    There are some doubts about the results presented in Figure 3, as follows: (1) Figure 3 (A) The Y-error is quite large when the glucose concentration is 6, 9, and 15 mM, which will cause serious problems in quantitative detection? In addition, glucose The capacitive response approaches 0 at a concentration of 30 mM, should this concentration not be included in the calculation? (2) Figure 3 (C) The capacitive response of the SRE sensor without hydrogel to glucose concentrations at 3, 6, 15, and 21 mM is almost at the same response point, and the glucose concentration at 9 mM almost approaches 0? (3) Fig. 3 (D) In the recording of the complete response of time to capacitance, the capacitance response decreased greatly when the glucose concentration was 3 and 6 mM. Why is it not proportionally decreased? The authors should state this in the Results and Discussion.

5.    In the long-term stability test of the sensor, is the DexG-Con A hydrogel sensor stored at room temperature or under refrigeration? In addition, are any reagents added during the storage process? And whether the author has conducted 2, 3, 4 weeks of testing, will the drift rate be so good? Because the general sensor is often stored for a long time, the response signal declines and the drift rate increases, which affects the stability of the sensor. Please state the author in this section.

6.    What is the pH value in the process of incubating cells with DMEM medium? Because the pH value of the general medium is suitable for a range of 7.2 to 7.4, but some special cells like a slightly acidic or alkaline environment, it may be necessary to adjust the pH value to 7.0 or above 7.4. The author does not explain in the text, please add in the experiments and results and discussions.
